# Superheroes or Super Spreaders? The Impact of the COVID-19 Pandemic on Social Attitudes towards Nurses: A Qualitative Study from Poland [note 1]

**DOI:** 10.3390/ijerph20042912

**Published:** 2023-02-07

**Authors:** Katarzyna Wałowska, Jan Domaradzki

**Affiliations:** 1Department of Gastroenterology, Metabolic Diseases, Internal Medicine and Dietetics, Heliodor Swiecicki Clinical Hospital, Poznan University of Medical Sciences, 60-355 Poznan, Poland; 2Laboratory of Health Sociology and Social Pathology, Department of Social Sciences and Humanities, Poznań University of Medical Sciences, 60-806 Poznań, Poland

**Keywords:** attitudes, COVID-19, experiences, nursing personnel, Poland, public perception, SARS-CoV-2, social image

## Abstract

The social perception of the nursing profession in Poland is profoundly affected by social stereotypes that may discourage young people from entering the profession and lead to prejudice towards nurses. During the COVID-19 pandemic, nurses gained visibility, which caused the social image of nurses to likewise grow. In this study, we explore nurses’ experiences with respect to how the COVID-19 pandemic influenced the social image of nursing. Semi-structured interviews were conducted with fifteen hospital nurses. Three major themes emerged: (1) social attitudes towards nurses during the pandemic, (2) nurses’ perception of the impact of the pandemic on the social image of the nursing profession and (3) the impact of the pandemic on nurses’ mental health. Although the pandemic promoted the image of nursing to the general public, nurses were disappointed that in the face of the healthcare crisis and the constant threat, they had to struggle with difficult working conditions and professional, social and economic recognition. This study therefore stresses the duty of policy makers to undertake a systemic approach to improving the organisation of health care and increase nurses’ safety by providing them with a safe working environment and prepare them better for the next health crisis.

## 1. Introduction

The coronavirus disease of 2019 (COVID-19) swept rapidly across the world in 2020, infecting more than 600 million people, and was associated with more than 6 million deaths by the end of 2022 [1]. The pandemic has therefore fundamentally challenged healthcare systems worldwide and has tested their resilience [2]. Whereas the outbreak has indicated the weakest points of health systems, it has also demonstrated the vital role healthcare professionals (HCPs) play during times of disaster in ensuring a functioning healthcare system and healthy society [3,4]. The pandemic has also highlighted the physical risks and the high levels of psychological stress HCPs experience at work (extremely demanding environments, long working hours, living in constant fear of exposure to the disease, separation from families, social stigmatisation, etc.) [5].

The first case of infection with the recent severe acute respiratory syndrome coronavirus 2 (SARS-CoV-2) in Poland was confirmed by the Polish Minister of Health on 4 March 2020 [6]. Whereas at the beginning of the pandemic, the daily number of infections in Poland remained in the dozens, according to the Polish legal act on the spread of infection, a state of natural disaster was soon declared [7], and on 10 March, the Polish government imposed a number of public health restrictions and lockdown-type control measures. This included the implementation of social distancing; limits on travel; and the closure of all schools, public places, hospitals and long-term care facilities to external visitors. In spite of this, the first death caused by the new virus occurred on 12 March [8], and on 20 March, a state of epidemic had been declared in Poland [9].

As in other counties, the pandemic created a situation in which nursing personnel had to operate under conditions that represented a threat to their health and influenced their teamwork and the climate of safety, job satisfaction, the perception of management and working conditions and their mental health [5,10,11]. The COVID-19 pandemic also resulted in great suffering among HCPs, including nurses, as since the beginning of the outbreak, more than 110,641 nurses became infected in Poland, and more than 260 died [12].

Although the pandemic caused chaos in the provision of nursing care in terms of working conditions (disinfection, overalls, masks and shields, gloves, goggles and PPE) and made nursing work even more exhausting both physically and emotionally, it also created an opportunity to elevate the public image of the nursing profession [13,14,15]. The reason for this is that over past two years, the critical roles and responsibilities of nurses during the pandemic expanded dramatically and came under increased scrutiny. While nurses continued in their roles on the front line of patient care in hospitals and other healthcare facilities, they were also actively involved in the screening, early diagnosis and continued monitoring of infected patients. They documented patients’ health status and communicated it to other health officials. Finally, nurses were responsible for informing patients about vaccinations and sometimes had to assume the duties of other personnel, i.e., technologists [3,4].

Particularly during the first wave of the pandemic and as a result of governmental efforts, media coverage and a number of social campaigns, nurses received a great deal of social attention, highlighting their sacrifice, bravery, tremendous efforts, dedication and altruism. This, in turn, increased the visibility of nurses, elicited recognition for their work during the crisis and highlighted the reason the role of nurses is so important in health care, thereby promoting the image of nursing to the general public.

Polish legal regulations define nursing personnel as independent professionals [16,17]. Nurses’ independence and autonomy are simultaneously based on social trust [18,19]. While the average number of nurses per 1000 population in the European Union (UE) is 8.8, in Poland, it is 5.1—one of the lowest proportions in all OECD countries. In Switzerland, it is 18.0; in Norway, it is 17.9; in Iceland, it is 15.4; and in Germany, it is 13.9 [20]. While there are currently 232,387 nurses working in Poland, according to the Main Chamber of Nurses and Midwives, this number is expected to decrease by 16,761 by the year 2025 and by 36,293 by 2030 [21]. Finally, while the average nurse in Poland is 53.71 years of age, more than one-third of all Polish nurses is between 51 and 60 (35.8%) years of age, and 27.2% are aged over 61 years.

Although in order to improve the status of nursing worldwide and empower nurses, the World Health Organization (WHO) and the International Council of Nurses launched the *Nursing Now Campaign* (2018–2020) and the World Health Assembly designated the year 2020 the *Year of the Nurse and the Midwife* [22], the nursing profession is becoming progressively less popular in Poland. Research conducted among secondary school students demonstrated that while young people in Poland associate the nursing profession with helping people and a high degree of commitment and diligence, they also believe it to be a low-paying profession with low social prestige and too much responsibility. Therefore, very few students declare an interest in becoming a nurse [23,24].

This social perception of the nursing profession, as defined as a process in which individuals make certain judgements or draw inferences about individuals or groups [25], is profoundly affected by stereotypes and is formed through discourses of knowledge and power [26,27] that shape the public understanding of what is considered to be true and false. Indeed, research shows that in Poland, nursing is still perceived as a woman’s job, and nurses are often imagined according to traditional, sentimental stereotypes as selfless young women [26,28,29]. Moreover, for historical and cultural reasons, they are often ascribed a secondary role and are perceived as lacking skills and training, with little responsibility, autonomy or decision-making capacity [23,24,28,30]. Some also believe that nurses are failed medical students, i.e., that they are insufficiently intellectually equipped to become doctors [24,28]. This is because the nature of nursing work is often seen as simply caring for and helping patients. Consequently, nurses’ work is perceived as inferior to that of doctors, and nurses are perceived as doctors’ assistants or servants [23,24,28]. On the other hand, male nurses are often labelled as unmanly or gay and have to challenge the stereotypical view that they are unsuitable caregivers [26,29].

However, these stereotypes are also present in other countries. Numerus studies report that while nurses are considered kind, caring and hardworking, they are also frequently perceived as less intellectual or autonomous than other professionals [31,32,33]. Moreover, people often believe that nurses’ training is easy and that all hospital nurses do the same kind of work, i.e., provide care for the sick in hospital, administer medication and deal with patients’ hygiene [31,32,33,34]. On the other hand, even if nurses are perceived as responsible, skilful, compassionate and kind, people are unaware of such important competences as research, leadership, involvement in decision making, teamwork competences and interprofessional communication, which are necessary in modern nursing [31,33]. In summary, research suggests that nursing is frequently imagined as simple, blue-collar work offering low social prestige [31,32,33,34,35].

This is important because such harmful stereotypes tend to discourage young people from the profession and may give rise to prejudice towards nurses [30]. They may also affect nurses’ mental health and lead to frustration, low self-esteem and a disruption of nurses’ sense of worth, in addition to making their job considerably more difficult [33,34,35,36]. Although in a recent public poll, nursing ranked second on a list of the most respected professions in Poland (89% of responses), just behind fireman (94%) and ahead of doctors (ranked sixth, with 80% of responses) [37], Radosz et al. demonstrated that the social image of the nursing profession in Poland is poorer that in France and Portugal [38]. While only 14% of Poles believed nurses’ professional status among the medical professions to be high, 50% believed it to be mediocre and 26% to be low. Poles rarely knew that a nurse could undertake a medical specialisation; they tended to assign nurses a subordinate position and stress nurses’ caring, therapeutic role insofar as they follow doctors’ orders; and they believed nursing to be a poorly paid profession. Many stressed that the media perpetuate the social image of a nurse as a cheap sex symbol.

Of equal importance is that nurses themselves tend to perceive their professional status as average (49.7%) or low (25.4%) [39,40]. Studies also show that during the past decade, the evaluation of nurses’ professional prestige among health professions has been poor, largely among doctors and nurses [36,41]. Polish nursing students also perceive the social prestige of their putative future profession as mediocre (32.3%) or low (31.7%), and their image of nurses is profoundly influenced by cultural stereotypes, i.e., as a well-made-up middle-aged woman [35].

Furthermore, as the social image of nurses has changed over the decades from that of a servant to patients and subordinate to doctors to a professional caregiver [30]; some argue that the social image of nurses may be strengthened during times of disasters such as earthquakes, tsunamis and epidemics [13]. It is also suggested that the SARS-CoV-2 healthcare crisis has clearly stressed the status of the nursing profession and the vital role nurses play in promoting public health and that COVID-19 has provided a unique opportunity to strengthen the public image of nurses and extend their potential to influence health policy and decision making at both national and global levels [14,15].

In this study, we explored nurses’ experiences with respect to how the COVID-19 pandemic influenced the social image of nursing by asking the following research questions:What social reactions did nurses face during the pandemic?Has the pandemic changed the social perception of nurses?

## 2. Materials and Methods

### 2.1. Study Design

While earlier studies have focused on the experiences and attitudes of HCPs towards the COVID-19 outbreak or the impact the pandemic has had on their mental health, there remains a shortage of research on the impact of the pandemic on the social perception of nurses.

Given the scarcity of research on the impact of the COVID-19 on the social perception of nurses in Poland and because research rarely gives voice to nurses themselves and their lived experiences, this research was designed as a qualitative study [42,43]. Semi-structured interviews were conducted with nurses working in hospitals during the COVID-19 pandemic in Poznan, Poland. After a thorough analysis of academic literature, the interview schedule was designed to ascertain the experiences of nurses’ on hospital wards during the coronavirus crisis and the way these experiences influenced their decisions and choices. The participants were interviewed about their personal experience on hospital wards during the pandemic, including the reactions they encountered as nurses in hospitals and their perception of the impact of the COVID-19 pandemic on social attitudes towards nurses. The interview schedule consisted of 12 questions intended to facilitate the identification of specific issues related to nurses’ experiences during the COVID-19 crisis.

Before the formal phase of the qualitative research, three pretest interviews were conducted with nurses to assess the rigor of the instrument and to formulate measures to address any limitations or threats to bias and management procedures. This resulted in a reformulation of four questions. The final version of the interview schedule was evaluated by two external reviewers: a nurse and a sociologist.

In order to explore nurses’ experiences during the COVID-19 pandemic and their perception of its impact on the social perception of nurses, an interpretative phenomenology approach (IPA) was used [44,45]. This approach was selected because, as IPA is participant-oriented, it offers researchers the opportunity to determine and understand the way participants make sense of their experiences. At every stage of the interviews, nurses were therefore encouraged to express themselves freely when reflecting on their professional lives during the COVID-19 pandemic. This design permitted respondents to provide new information about their experiences and what they mean. This, in turn, generated in-depth knowledge regarding nurses’ perceptions of the impact of the COVID-19 pandemic on social attitudes towards nurses.

### 2.2. Participant Recruitment and Data Collection

The participants were chosen among nurses who worked in Polish hospitals, were directly involved in hospital care during the COVID-19 pandemic and were willing to participate in the study.

Participants were identified via an advertisement that was posted on an online platform for nurses. Nine hospital nurses initially responded to this advertisement and volunteered to participate. However, in order ensure the validity of the results, recruitment of nurses continued until thematic saturation occurred. To this end, a non-probability snowball sampling method was also used [46]. After making personal contact with the first author (K.W.), nurses who were scheduled to be interviewed were asked to provide referrals to other potential subjects. This helped to recruit six additional nurses. The recruitment process continued as such until thematic saturation was achieved [47]. Ultimately, a total of fifteen nurses responded and agreed to an interview.

Nurses were interviewed between February and May 2022. A total of 10 interviews were conducted in person, and 5 were conducted via telephone conversations lasting between 25 and 45 min. All interviews were digitally audio-taped and transcribed verbatim, noting intonation, emphases and emotions.

### 2.3. Ethical Issues

This study was performed in accordance with the principles of the Declaration of Helsinki. In accordance with Polish law and good clinical practice for research involving human subjects, this study required no revision by an ethics committee.

Nurses received an invitation letter and were informed about the study’s purpose, as well as the voluntary, anonymous and confidential nature of the study and about the possibility of withdrawing from the interview at any given moment and/or not disclosing information regarding personal circumstances. Respondents were also informed that identifying information would be redacted from records of the interviews and that they would be stored in a secure place with restricted access. Participants were informed that, should they experience any emotional distress while recalling past events, a hospital psychologist was available for counselling and that they could take time to collect themselves and/or decide whether to continue the interview.

After informed consent was obtained from all respondents enrolled in the study, they were scheduled to be interviewed. To ensure that the participants were psychologically safe and anonymous, all interviews took place in private, isolated rooms [48]. All interviews ended without any requests for further assistance.

### 2.4. Data Analysis

Once the interviews were collected and digitally audio-taped, all transcripts were anonymised and read multiple times for coding purposes, then categorised during familiarisation. Key words, significant phrases and statements describing nurses’ experiences during the pandemic were then sought. The initial results were assigned preliminary codes with the relevant supporting quotes selected [49]. They were then consolidated into meaning statements, which were grouped according to themes and subthemes. Any conflicting data were discussed, and resolutions were reached within the team. Once agreement had been reached, the final categories were read and analysed using thematic analysis of the transcripts. Because all interviews were conducted in Polish, all quotes and themes were translated into English during data analysis and reporting with the help of bilingual translators [50].

## 3. Results

### 3.1. Participants

A total of 15 nurses, all of Polish origin, were interviewed: 13 women and 2 men (mean age: 46.67; range: 23–65; mean years of experience: 20.8; range: 2–45) (Table 1). Eight respondents held master’s degrees in nursing; two had bachelor’s degrees; three had graduated from medical high school; and two held PhD degrees. While the majority were unspecialised nurses, seven had specialisations: in oncology (*n* = 2), in diabetes (*n* = 2), in surgery (*n* = 1), in internal medicine (*n* = 1) or in conservative medicine (*n* = 1). Six respondents worked as full-time nurses, five worked 80 h weeks with two jobs and four worked 12 h shifts.

### 3.2. Findings

Three key themes emerged from the interviews: (1) social attitudes towards nurses during the pandemic, (2) nurses’ perception of the impact of the pandemic on the social image of the nursing profession and (3) the impact of the pandemic on nurses’ mental health (Figure 1).

#### 3.2.1. Social Attitudes towards Nurses during the Pandemic

##### Nurses as Heroes: Pride, Respect and Admiration

In considering the way nurses were perceived by others and the reactions they faced during the pandemic, most respondents mentioned encouraging feedback from their families, relatives, neighbours and friends, including support, appreciation, pride and respect. Some also experienced acts of kindness and gratitude from their patients, who thanked them for their care. The general public also acknowledged nurses for their service and involvement in the fight against the pandemic. Such reactions were important because they provided nurses with extra motivation and helped them to cope with stress and difficult working conditions.


*I will never forget the moment an older woman, the same age as my grandmother, thanked me for every word, every gesture toward her. When she recovered and was discharged, she gave me her address and invited me for a delicious dessert because she wanted to thank me. I felt such gratitude and appreciation, and it has helped me to manage all these emotions and all the hard work.*
(N8)

Respondents acknowledged that although COVID-19 has made nursing exhausting, both physically and emotionally, it has also increased the social visibility of nurses, as they are more recognised for their professionalism, skills, commitment and dedication. As people recognised the extent to which they were essential to the healthcare system, nurses were also admired as heroes.


*It was nice when my neighbour, an older lady, came to me with a cake and said that without us there would be nothing. I was really moved by that.*
(N10)


*I was working in a temporary hospital at Poznań International Fair. Once an older patient who was in a really bad state grabbed my hand and said, “My child, thank you from the bottom of my heart. If not for you, I would be long gone”. Such words gave me strength, even though there were moments when I wanted to leave it all and just rest.*
(N15)

##### Ambivalence: Between Indifference and Suspicion

Respondents admitted that at the beginning of the pandemic, some people, including their families and relatives, were ambivalent towards nurses. While they acknowledged the importance of their service and appreciated their dedication and sacrifice, they were also afraid, attempted to discourage them and suggested that working in dedicated COVID-19 hospitals was irresponsible, as they might pose a risk to others.


*My grandmother was constantly saying, “My child, let it go, you are young, you have your whole life ahead of you. They will manage without you. You have a family, a child, a husband”.*
(N6)

*At the beginning my relatives were more afraid of me than the pandemic itself* [laughs]. *They avoided contact with me. We even spent Easter apart. And the higher the number of infections, the greater their fear.*(N7)

##### Nurses as Frauds: Prejudice, Stigmatisation and Discrimination

However, many nurses emphasised that during the first wave of the pandemic in particular, they faced many upsetting reactions, prejudice, stigmatisation and even discrimination, as people “were very much against us” (N11). According to respondents, such reactions resulted from the fact that many people associated nurses with the restrictions imposed by the government, especially on entering hospitals and public health requirements, i.e., to wear masks, to socially distance and to limit hospital visits. As nurses symbolised limits on people’s personal freedoms, respondents admitted to facing offensive or aggressive behaviour, including insults. All respondents stressed that such distressing reactions and discriminatory behaviour towards medical staff upset them and were disappointing and emotionally challenging.


*I felt very sad when I heard in a shop that medics and pensioners are privileged because we could do our shopping without queueing. One lady called us “parasites”. I did not even enter into a discussion because I was afraid that those people would point at me.*
(N7)

Nurses were also frustrated at being treated as part of a conspiracy. From the very beginning some people believed that the virus was a hoax, and this led to nurses being accused of being one of the groups behind the pandemic.


*From the very beginning my uncle was very sceptical. He sniffed out a conspiracy and argued that the medics are frauds, not heroes. That hurt a lot, especially when I came home from twenty-four hour shifts, filled with the chaos of people dying in our arms, and others seeking out a conspiracy.*
(N3)


*When my sister-in-law found out that I had been vaccinated, she told me that we couldn’t meet, as the vaccinations could affect her, as they contained heavy metals that would be deposited in the brain.*
(N12)

Some respondents also complained of being accused of making money. While some people argued that it was medical staff’s duty to make sacrifices because they knew what they had signed up for when they chose their profession, others suggested that nurses were public servants paid by tax revenue and that even those working on the COVID-19 front lines should do their jobs for no additional remuneration.


*The worst thing I have ever heard was that I was doing it all for the money!.*
(N2)


*One thing stuck in my memory: “How are you doing with the double salary paid from my taxes? There are more important expenditures! You should not get a penny for it. It’s all a scam!”.*
(N9)

Although some nurses suggested that the COVID-19 reporting in the media has helped raise awareness of the role of nurses in society, others stressed the destructive role of social media in disseminating suspicion and doubt towards nurses. While respondents complained about hatred on the Internet, they emphasised that such attitudes became more common as the pandemic progressed.


*There were many disparaging comments about us on Facebook, especially under daily reports on the infection rate published by the Ministry of Health. It could be depressing, as people were very cruel, even heartless toward nurses, doctors and paramedics.*
(N5)


*Some did not believe that people were really dying in the hospitals. Others thought that the image was created for TV. (…) I remember one such terse comment. One man wrote under the article: “They did not save my mother, I wish them the same”. Since then I have stopped reading comments.*
(N13)

#### 3.2.2. Nurses’ Perception of the Impact of the Pandemic on the Social Image of the Nursing Profession

##### Spotlight on Nursing

Some respondents claimed that the major way in which the COVID-19 pandemic affected the social image of nurses was that nurses acquired great social visibility. They experienced the sensation of being under constant scrutiny. Younger respondents and those who had just begun their nursing career in particular suggested that it made them feel uncomfortable, as every action of nurses was widely discussed and commented on.


*Yes, during that time we were in the public eye. Our every move was commented on, and every stumble was pointed out.*
(N13)

##### The Role of (Social) Media

Nurses stressed that before the pandemic, nurses were often associated with social protests or were perceived through gendered or historical stereotypes of female nurses, possibly religious nuns, who dispensed care and charity as part of their public service. While stressing this, respondents emphasised that such false and hurtful images are still depicted by TV series, films and medical dramas.

*Nurses are not nuns! Those times are long gone. Neither are we “piguła”* [Polish for pill, a pejorative term for nurses]. *I hate that term.*(N5)


*TV series have a huge influence on social respect towards nurses. I have often seen nurses depicted in skimpy outfits. Does it help? Of course not! We really have expertise, a great deal of responsibility, and the last thing we think of is parading about in short skirts, as some people seem to think.*
(N6)

##### Social Image of Nurses before COVID-19

Reflecting on the impact of the pandemic on the social image of nurses, respondents also referred to their previous experiences and what they had heard about their profession in the past. Most declared that nurses were often thought to be tired, older and uneducated women who need a doctor in order to carry out their jobs. All respondents were frustrated at being treated as public servants subordinate both to doctors and patients rather than as well-educated, highly skilled and independent professionals.

*POZ* [Basic Health Care] *is bad for our image, as patients often think that a nurse is simply a person who works in registration.*(N11)


*People do not know that in order to become a nurse you have to graduate from university. Once, when I was doing my practical training on a ward I was asked by a lady what I was doing there. When I told her that I was doing my practical training because I was studying nursing, she was very surprised. “Do you have to study for that? I thought that a regular course was enough”, she said.*
(N14)

##### Social Image of Nurses after COVID-19

However, many respondents argued that the pandemic raised social awareness of the essential nature of nurses’ role in the medical hierarchy. They also stressed the way the public recognised and acknowledged that the nursing profession requires special qualifications, knowledge, skills and internal motivation. As most nurses felt appreciated, important and needed, they believed that the COVID-19 outbreak had a beneficial influence on the visibility and perception of the nursing profession. Furthermore, respondents hoped that their profession would gain further esteem, recognition and recompense.

*For the very first time, after 25 years, I have heard somebody acknowledge my work. Just thinking about it brings tears to my eyes, as I hope that it* [the pandemic] *will finally change the image of Polish nurses.*(N5)


*When I was working on the COVID ward, during one shift a man of 36 thanked me for everything. He said that nurses were doing a great job and should be lauded for that. I talked with him for a moment and he admitted that his previous experiences were poor, but now he realised what it was like and he appreciated our work.*
(N12)


*People have finally realised that nursing is an independent and responsible profession, and not only one of “pass it; bring it” to a doctor”.*
(N7)


*Thanks to the pandemic our profession has become more independent (…). During one shift I heard a conversation between a nurse and a patient, who told her that the pandemic had made him realise how much work we have, how much we know and how little we are acknowledged.*
(N11)

##### COVID-19 and Nurses’ Professional Identity

It is equally important that, apart from bolstering the social image of the nursing profession, some respondents suggested that the pandemic has also improved nurses’ self-esteem, professional self-image and identity.

*In my opinion, it* [COVID-19] *had a beneficial influence on us. It has strengthened our belief that what we do is good, necessary and that without us the entire system would collapse.*(N3)


*It was hard to enter the profession at such a difficult time. It made tough and inured me a great deal. Now I simply do my thing.*
(N13)

#### 3.2.3. The Impact of the Pandemic on Nurses’ Mental Health

##### COIVD-19 and Disrupted Lives

All nurses emphasised that the COVID-19 pandemic created a new normal that seriously affected their mental and emotional well-being. While the inconveniences of lockdown and difficult working conditions exhausted them physically, living and working in a state of the permanent threat of infection or of transmitting the virus to their loved ones made them hypervigilant and prone to over-reaction and resulted in an increased level of anxiety, psychological distress and symptoms of depression. All these painful emotions disrupted their everyday lives.

*I have not known anxiety disorder before (…). I have never experienced it before. Me? No way. Such a strong woman! All my strength disappeared when* COVID *emerged around me, and then came death. How can you accept the death of a person who was the picture of health just few days before (…) That’s why I began to see a psychiatrist.*(N13)


*My mental health deteriorated, but I am no longer afraid to talk about it. However, in the beginning I didn’t want to admit that this whole situation overwhelmed me completely; and yes, I felt in danger, like I was standing next to a bomb that could explode at any second.*
(N1)

##### COIVD-19 and Disrupted Work

All respondents complained of staff shortages, inadequate personal protective equipment (PPE) and the everyday struggle with difficult working conditions. They frequently mentioned that, as many HCPs had to quarantine, those who were healthy were burdened with additional duties and felt a great responsibility for additional patients. This, in turn, resulted in many feeling that they were stretching themselves at work beyond their capacities and had no time for physical or mental regeneration. Nurses also complained that they were denied legitimate leave. As many respondents were working 80 h per week, they were often at the hospital for more than 48 h without seeing their families. They also lived in a constant state of uncertainty.


*Apart from the fear about the health of my relatives I was burdened by the awareness that I was the head of the family and have to bring in the money (…) My wife was pregnant back then and I felt a great responsibility.*
(N2)


*Do you know the feeling that you have the weight of the world on your shoulders? There was a moment on my ward, when, except for me, only two nurses were healthy and not quarantined. It was a nightmare. I thought that I would soon be next. We were alone during the shifts, working 24 h. And back then it was my first job. What is even worse, I had just been working for five months. I always brought a big bag full of things with me, as you didn’t know what would happen that day.*
(N14)

Some nurses complained that, apart from long shifts at work, they were exhausted, both physically and emotionally, by working in PPE (overalls, masks and shields, gloves and protective glasses). Others mentioned that during the first wave of the pandemic in particular, they felt anxious about the lack of PPE, as there were very few masks, visors and disposable aprons.

The restrictions enforced by the policy regulations on the working of hospitals constituted another source of stress for nurses. Respondents emphasised that, both as HCPs and family members, they were stressed that, as Primary Healthcare (POZ) was closed, even the simplest healthcare services that might easily be addressed by family doctors were available only in Hospital Emergency Rooms (SOR). Nurses therefore emphasised that this organisation of the healthcare system not only caused them more work but also resulted in prevalent irresponsible behaviour among patients, who often concealed infection, even when they experienced symptoms, thereby creating additional stress.


*I work in the emergency unit and it affected us a lot. They called us to literally everything, but people would not tell us when they were infected.*
(N9)


*Why did they want to protect doctors and nurses working in Primary Healthcare (POZ), but not us?.*
(N13)

##### COIVD-19 and Emotional Distress

The risk of infection or of spreading the virus to their families, as well as difficult working conditions, resulted in emotional distress among nurses. Some nurses consequently experienced difficulties in keeping their emotions in check and described their feelings of frustration, irritation and anger. This, in turn, affected their relationships with their partners, their children and their friends. Some said that their children also became nervous and that the pandemic had totally dominated family conversations and time spent together, which resulted in relentless fear mongering regarding COVID-19. Nurses also stressed that with every new wave of infections and deaths, these emotions increased and made them feel that the crisis would never end. Every time restrictions were relaxed, nurses were worried that the pandemic would soon strike again with redoubled force. Respondents also stressed that this fear was constantly fuelled by the media’s creation of an omnipresent image of an apocalypse.


*Eight months of constant stress, watching all the news. The wave of distressing information pounded us. I could not sit still and grew nervous more easily. I felt like a ticking bomb. And this stress infected my children.*
(N6)


*I was seriously stressed by constantly watching the news: how many people were infected every day; how many had died, etc. As these numbers increased my anxiety grew proportionally.*
(N12)

## 4. Discussion

This study supports findings from other studies that showed that the COVID-19 pandemic provoked some important changes in the social image of nurses. For example, most Spanish patients appreciated the humanised nursing care received during the pandemic in terms of the quality of nursing practice, nurses’ willingness to care and their openness to nurse–patient communication [51]. A Korean study demonstrated that as a result of the pandemic, the role of nursing staff in the prevention and management of mass infections has been recognised by the public [52]. Almost 86% of nursing students in China believed that COVID-19 had a beneficial influence on the image of nursing [53].

Because nurses constantly faced the risk of contracting the virus at work, the pandemic damaged respondents’ relationships with others, including families and relatives, as well as with patients and other HCPs [54]. This study confirms that although nurses have often been heralded as heroes, the global fear and panic caused by the COVID-19 outbreak resulted in increased prejudice, stigmatisation and discrimination against HCPs [55,56,57,58]. Studies conducted in Indonesia [59], China [53] and Italy [60] also demonstrated that especially during the first wave of the pandemic, nurses faced stigmatisation resulting from accusations that nurses spread the pandemic and were delinquent in their duty of care. However, this study suggests that as the pandemic progressed, such upsetting reactions also resulted from the fact that nurses were often perceived as government officials who imposed COVID-19 restrictions and frustrating preventative measures on society [61].

Most importantly, this research is in line with findings from other studies that have emphasised that while nurses have made gains in terms of social recognition and respect, many respondents were disappointed that in the face of the healthcare crisis and constant threat, they had to struggle not only with difficult working conditions but also professional, social and economic recognition [62,63,64]. Indeed, the nurses enrolled in this study stressed that although they were recognised by the public as essential workers during the pandemic and the media often framed them as heroes or angels, they did not always find such images helpful, as their profession was still devalued and undervalued. They also hoped that their efforts and sacrifice would not be forgotten and would translate into an improvement in nurses’ working conditions. A study by Stokes-Parish et al. showed that critical care nurses rejected the media narrative that described nurses either as ‘heroes’ or ‘angels’ [62,64,65]. Similarly, although Italian nurses felt appreciated by the public, they also felt devalued and unrecognised as professionals [61]. Rossi also demonstrated that Italian nurses failed to recognise themselves in the images in the mass media and published on social networks stereotyping them as icons of heroes. They also stressed that while such an image captures the attention of the public, it fails to emphasise the skills the nursing profession demands [63].

Some respondents were also frustrated that, even during the pandemic, the media perpetuated gendered stereotypes of nurses without emphasising their professional knowledge and skills, thereby disempowered the nursing profession. However, other studies have argued that the social image of nurses as self-sacrificing heroes masks the requirement for extensive education; improvement in working conditions; and professional, social and financial recognition [62,63,64,65,66]. Some authors also argued that, as such a narrative acknowledges nurses but ignores other healthcare professionals, it reinforces the biomedical model of care and threatens person-centred care [58]. Others suggested that the heroic narrative may be damaging because it stifles social discussion of role-related conflicts [67] and the limits of HCPs’ duty of care during the pandemic and the importance of reciprocity [68].

Another important finding is that some respondents declared that nurses were insufficiently protected (lack of PPE) and experienced work overload (difficult working conditions, staff shortages and other obligations outside their jobs). They consequently felt they had fallen victim to government negligence. Bennett et al. also argued that glorification of nurses shifted responsibility from the government’s inability to devise a plan for the healthcare crisis for individual healthcare workers [15]. Mohammed et al. go even further and suggest that ’the hero discourse’ is a political and ideological tool that helps to normalise nurses’ exposure to risk and dangerous occupational conditions, which are constructed and justified as inevitable in nurses’ professional life. They also argue that such rhetoric limits nurses’ ability to determine the conditions in which they work and therefore helps to preserve existing power relationships [27]. Previous studies conducted among Polish medical student volunteers also demonstrated that they felt unprepared for work during the pandemic and lacked social skills and access to psychological support. While many felt unsafe at work, some felt they were being exploited [69].

## 5. Implications for Policy and Practice

As all respondents highlighted personal risk, lack of adequate PPE, shortages of skilled staff and low nurse-to-patient ratios (particularly in critical care), this study also suggests that social admiration for nurses should not distract our attention from decades of neglecting health services, failing to prepare adequately for epidemic crises and neglecting to foster the social prestige of the nursing profession. The reported results therefore stress that society and policy makers have a duty to undertake a systematic approach to improve the organisation of healthcare, increase nurses’ safety by providing them with a safe working environment, better prepare them for the next health crisis and reward nurses for their sacrifices.

While the COVID-19 outbreak has helped raise social awareness of the role of nurses in the healthcare system, there is an urgent need for public communication and the promotion of a more truthful image of nursing in society.As working during the outbreak was physically and emotionally challenging, special attention must be paid to HCPs’ safety, and they must be provided with both the necessary personal protective equipment and mental health support.Nurses and HCPs in general should be better prepared for future global health crises and medical disasters, emergency decision making, coping and leadership during a crisis by medical curricula.

## 6. Limitations

Although to the best of our knowledge, this is the first study to explore nurses’ experiences during the COVID-19 pandemic in Poland, it is not without limitations. First, although thematic saturation was achieved, the study sample was relatively small, with only fifteen nurses interviewed. Secondly, because nurses from only one Polish city were interviewed, this study is geographically limited. Thirdly, female participants predominated (F:M 13:2), which limits the transferability of the results to male nurses. This imbalance is nevertheless representative of the female predominance within the nursing profession in Poland. A further limitation is that only one method of data collection was used without triangulating it against other sources, i.e., patient interviews or media discourse analysis. The data analysis might also be considered a limitation, as researchers did not code the transcripts independently but collaborated to decide on codes. Finally, because this study included non-English-speaking participants, all interviews were conducted in Polish, and all quotes and themes were translated to English with the help of bilingual translators. Because every language reflects important nuances that are inherent in participants’ experiences and situational context, translation of participant quotes from one language to another may disrupt theses nuances, and some meanings may be lost. Every translation involves the risk of misinterpretation on the part of researchers [70].

Despite these limitations, the benefits of this study should also be acknowledged. Most importantly, because there is a limited amount of prior research on the topic, this study narrows the gap in the literature regarding the experiences of nurses during the COVID-19 pandemic in Poland.

## 7. Conclusions

This study shows that although most respondents experienced positive reactions and social gratitude for their service and involvement in the fight against the pandemic, many nurses faced prejudice, stigmatisation or discrimination, which made them feel undervalued.

Our findings also confirm that the COVID-19 outbreak created an opportunity to strengthen the social image of nurses. Many participants believed that the pandemic has helped to overcome harmful stereotypes of nurses and disseminate the image of nurses as hard-working, well-educated, highly skilled and independent professionals. This, in turn, improved respondents’ self-esteem and professional identity.

However, this study also shows that because the government, the media and patients often highlighted the gendered stereotype of a nurse as a caring and empathetic professional, the pandemic might have further undermined nurses’ professionalism and diverted social attention from the knowledge, education and skills required to become a competent nurse. While all nurses enrolled in this study acknowledged that the pandemic increased the social visibility of nurses, who have been recognised as essential healthcare workers, they were also frustrated that in times of healthcare disasters, they still had to struggle for social, professional and economic recognition.

## Figures and Tables

**Figure 1 ijerph-20-02912-f001:**
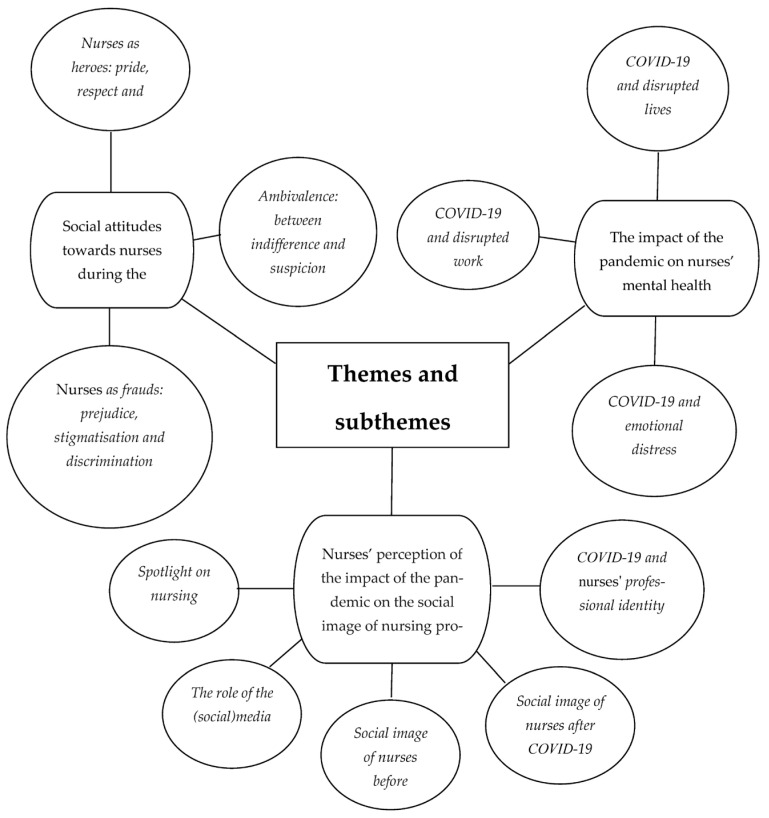
Themes and subthemes emerging from qualitative analysis.

**Table 1 ijerph-20-02912-t001:** Study participants.

Code	Gender	Age Range	Education	Specialisation	Seniority (in Years)	Hospital	Type of Employment
N1	Female	35–39	Master’s	Surgical	17	Private	60 h weeks
N2	Male	30–34	Master’s	-	5	Public	80 h weeks
N3	Female	25–29	Master’s	-	6	Public	40 h weeks
N4	Female	65–69	Medical high school	-	45	Public	40 h weeks
N5	Female	45–49	PhD	Diabetes	25	Public	20 h weeks
N6	Female	45–49	Fedical high school	-	27	Public	60 h weeks
N7	Female	55–59	PhD	Oncology	34	Public	40 h weeks
N8	Female	18–24	Master’s	-	2	Public	80 h weeks
N9	Male	55–59	Master’s	Oncology	34	Public	40 h weeks
N10	Female	50–54	Bachelor’s	Conservative medicine	31	Public	40 h weeks
N11	Female	55–59	Medical high school	-	37	Public	80 h weeks
N12	Female	45–49	Master’s	Internal medicine	22	Public	20 h weeks
N13	Female	45–49	Bachelor’s	Diabetes	22	Public	40 h weeks
N14	Female	18–24	Master’s	-	2	Public	80 h weeks
N15	Female	18–24	Master’s	-	3	Public	80 h weeks

## Data Availability

The data and codes that produced the findings reported in this article are available from the corresponding author upon reasonable request.

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
