# Peer review of "Superheroes or Super Spreaders? The Impact of the COVID-19 Pandemic on Social Attitudes towards Nurses: A Qualitative Study from Polandâ€"

_ijerph, 2023, doi:10.3390/ijerph20042912_

Round 1
Reviewer 1 Report
I enjoyed reading your article on an interesting topic. I noted the following issues which need to be resolved.
- My main issue is with how you use the concept of 'social perceptions' throughout the article. In the introduction section I noted a lack of evidence (referencing) when referring to the 'social perception of the nursing profession' (this is the top of page 2). Where has these suggestions come from? Previous literature on the history of nursing has discussed these 'stereotypes' and you need to refer to this. Also can the term 'social perception' be defined? Whose 'social perception' are we referring to? Societal perceptions? I would suggest referring to how societal perceptions have been formed through discourses of power and knowledge as Mohammed et al., (2021) discuss.
- I think the aim of the study (bottom of page 2) needs to be spelled out in a less confusing manner: '...their perspectives on the impact of the pandemic on the social perception of NPs ...' . Do you want data on how the public view NPs or data on how NPs think they are viewed? This is not clear from this statement. Likewise, I am not sure interviews with NPs will answer your 4 research questions. It will give you information on what NPs think about public attitudes towards them during the pandemic, but your interviews with NPs will not tell you if the COVID-19 pandemic has turned the 'spotlight' (define?) onto NPs, or if their 'social role has been socially recognised' (again what does this mean?). Therefore I think your study aims / research questions need refining in clear terms in light of what the data you have collected is able to tell you.
- I think the analysis would benefit with possibly less quotes and more analytic reflection on each quote.
- I think the discussion and conclusion sections are good.
Author Response
Dear Reviewer,
First of all, we would like to express our gratitude for giving us the opportunity to revise and resubmit our paper. We are indebted to you for your valuable suggestions and helpful comments. We hope that this revised paper is more consistent as a result of your willingness to help.
As you will see from this cover letter, we have put a good deal of effort into revising our paper in accordance with your suggestions. We are in agreement with the points you make and we are extremely grateful for pointing these things out and we believe that we have addressed all these points. Should we happen to have missed or misunderstood any vital comment, however, we would be more than happy to promptly rectify this and further revise the article.
Below we detail the changes we have made in accordance with your suggestions and comments. Please note that in the revised manuscript we have taken the liberty of marking the most important changes in colour to facilitate their checking.
At the same time, we would like assure you that, while prior to the original submission the manuscript has been read and copy-edited by a British native speaker, the revised paper has also been read and copy-edited by another British native speaker.
1. Reviewer’s comment:
I enjoyed reading your article on an interesting topic. I noted the following issues which need to be resolved.
Authors’ response:
Thank you. We appreciate your positive feedback and valuable suggestions on ways to improve our article.
2. Reviewer’s comment:
My main issue is with how you use the concept of 'social perceptions' throughout the article. In the introduction section I noted a lack of evidence (referencing) when referring to the 'social perception of the nursing profession' (this is the top of page 2). Where has these suggestions come from? Previous literature on the history of nursing has discussed these 'stereotypes' and you need to refer to this. Also can the term 'social perception' be defined? Whose 'social perception' are we referring to? Societal perceptions? I would suggest referring to how societal perceptions have been formed through discourses of power and knowledge as Mohammed et al., (2021) discuss.
Authors’ response:
Having been persuaded by your concerns, we have rewritten entire paragraph. While we have defined the term ‘social perception’ as the process where individuals are making certain opinions or inferences about individuals or groups [Aronson et al. 2010], we have also added suitable references to each stereotype described.
Additionally, in accord with your apt suggestion we have stressed that apart from stereotypes social perception of NPs has been formed through discourses of knowledge and power [Mohammed et al. 2021].
This has been done on pages 2-3 (lines 94-115). Thus, after revision it now says (the revised part has been marked in colour):
“This social perception of the nursing profession, as defined as a process in which individuals make certain judgements or draw inferences about individuals or groups [25], is profoundly affected by stereotypes, and is formed through a discourses of knowledge and power [26,27] that shapes the public understanding of what is considered to be true and false. Indeed, research shows that nursing is still perceived as a woman’s job and nurses are often imagined via traditional, sentimental stereotypes as selfless, young women [26,28]. They are also ascribed a secondary role and are perceived as lacking in skills and training, with little responsibility, autonomy or decision-making capacity [29-31]. Some also believe that nurses are thought to be failed medical students, i.e. that they are insufficiently intellectually equipped to become doctors [29,32]. This is because the nature of nursing work is often seen as simply caring for and helping patients. Consequently, nurses’ work is perceived as inferior to doctors’ and nurses may be described as doctors’ assistants or servants. Some people believe that nurses’ training is easy and that all hospital nurses do the same kind of work, i.e. provide care for the sick in hospital, administer medication and deal with patients’ hygiene [24,31,33]. While nurses are considered as kind, caring and hardworking they are therefore frequently perceived as less intellectual or autonomous professionals [33,34]. On the other hand, even if NPs are perceived as responsible, skilful, compassionate and kind, people are unaware of such important competences as research, leadership, involvement in decision-making, team work competences and inter-professional communication necessary to modern nursing [23,24,31]. All in all, research suggest that, nursing is frequently imagined as simple, blue-collar work offering low social prestige [23,24,31,32-35].”
We hope the above explanation satisfies your concerns. At the same time, again we appreciate your valuable suggestions.
3. Reviewer’s comment:
I think the aim of the study (bottom of page 2) needs to be spelled out in a less confusing manner: '...their perspectives on the impact of the pandemic on the social perception of NPs ...' . Do you want data on how the public view NPs or data on how NPs think they are viewed? This is not clear from this statement. Likewise, I am not sure interviews with NPs will answer your 4 research questions. It will give you information on what NPs think about public attitudes towards them during the pandemic, but your interviews with NPs will not tell you if the COVID-19 pandemic has turned the 'spotlight' (define?) onto NPs, or if their 'social role has been socially recognised' (again what does this mean?). Therefore I think your study aims / research questions need refining in clear terms in light of what the data you have collected is able to tell you.
Authors’ response:
We are grateful to for this remark, because it has helped us to clarify and revise the aim of our study. Thus, having been persuaded by your apt concern we have explained, that our research explored nursing personnel’s experiences in how the COVID-19 pandemic affected on the social image of nursing. Additionally, in accord with your suggestion we have reduced the number of research questions. This has been done on page 3 (lines 145-148). Thus, after revision it now says (the revised part has been marked in colour):
“This study explored NPs’ experiences in the way the COVID-19 pandemic influenced the social image of nursing by asking the following research questions:
- What social reactions did NPs face during the pandemic?
- Has the pandemic changed the social perception of NPs?”
We hope this change satisfies your concerns. At the same time, we are grateful to you for this valuable comment.
4. Reviewer’s comment:
I think the analysis would benefit with possibly less quotes and more analytic reflection on each quote.
Authors’ response:
In accord with your suggestion we have reduced the number of quotes to two per each thread. We have also developed analytic reflection on theses quotes.
Additionally we have divided each theme into sub-themes:
3.2.1. Social attitudes towards NPs during the pandemic (3.2.1.1. NPs as heroes: pride, respect and admiration; 3.2.1.2. Ambivalence: between indifference and suspicion; 3.2.1.3. NPs as frauds: prejudice, stigmatisation and discrimination),
3.2.2. NPs’ perception of the impact of the pandemic on the social image of nursing profession (3.2.2.1. Spotlight on nursing; 3.2.2.2. The role of the (social)media; 3.2.2.3. Social image of NPs before COVID-19; 3.2.2.4. Social image of NPs after COVID-19; 3.2.2.5. COVID-19 and NPs’ professional identity),
3.2.3. The impact of the pandemic on NPs’ mental health (3.2.3.1. COIVD-19 and disrupted lives; 3.2.3.2. COIVD-19 and disrupted work; 3.2.3.3. COIVD-19 and emotional distress)
We hope this change satisfies your concerns. At the same time, we are grateful to you for this valuable comment.
At the same time, we would like to point out that because the second reviewer suggested that Theme 1: “NPs’ reactions to the COVID-19 pandemic” is outside of the domain of the key research question about public views of NPs in the COVID pandemic and does not fit within the scope of what the paper, we have removed it both from the paper and discussion, and focused on the nurses experiences in how the COVID-19 pandemic influenced the social image of nursing.
Finally, we also wish to explain that the second reviewer’s suggested that Theme 2 “The impact of the pandemic on NPs’ mental health” should be placed as the last theme as it transits nicely into discussion on the challenges NPs faced during the pandemic and what needs to be addressed properly.
5. Reviewer’s comment:
I think the discussion and conclusion sections are good.
Authors’ response:
Again we wish to thank you for all your valuable suggestions and willingness to help us in improving our manuscript.

Reviewer 2 Report
Thank you for the opportunity to review this manuscript, which uses a qualitative approach to study the experiences and attitudes of nurse practitioners (NPs) in Poland during the COVID-19 pandemic. The paper talks in particular about NPs experiences and perceptions of societal attitudes towards them, and makes recommendations based on their lived experiences and responses for policymakers to improve their quality of working life.
This paper has the potential to make an important contribution to workforce literature, with nurse practitioners a key but understudied group, and Polish healthcare workforce of interest globally, and the examination of societal attitudes is a rich and useful area of focus given the discourse globally about the “healthcare hero” juxtaposed with public and political ‘blindness’ to the working conditions and psychological safety of healthcare workers. The authors suggest that there is a “great shortage of research on the impact of the pandemic on the social perception of NPs…in Poland” (line 102-3) but arguably there has been little examination of this anywhere, and since there are significant post-COVID pressures on healthcare systems and accompanying political and public rhetoric about healthcare workers and their needs and roles around the world (consider public and media discourse around nurse strikes in the UK, Australia and Ireland, rallies in New Zealand, doctors strikes in Malaysia Nigeria and Bolivia in the last 2 years alone), then showing how COVID changed public opinions of NPs in Poland has much broader international insights to provide.
However, I do think this piece needs substantial work to bring it up to a quality that would be suitable for international scholarly publication. I think a major issue is that the paper does not reflect a robust and well-understood qualitative research practice, and it perhaps tries to focus on too much, with the result that the core focus question of “how did COVID affect people’s perceptions of NPs, how did that affect them, and how do we fix it” gets lost. Re-establishing this core focus (or clarifying if this is not the right interpretation of the study!) throughout the introduction, findings, discussion is required, and the whole piece requires reframing in terms of cementing the scholarly qualitative epistemology and practice that underpins it to promote its robustness. It is also vital that the authors revise substantially their ethical research approach at the publication level, as there are ethical concerns raised regarding the data this paper shares and protection of the participants.
Specific comments:
Does the introduction provide sufficient background and include all relevant references?
I would suggest opening with a general statement about the importance of the healthcare workforce in the COVID-19 pandemic globally, and providing some basic facts/timeline of the pandemic in Poland in particular to set the scene – this is in the Discussion but needs to be moved much earlier.
Then focus in on the NPs – what was their crucial role in this? (This is later in the paper, lines 82-90, but needs to come earlier). It would also be helpful to define and detail what constitutes a “nurse practitioner” in the Polish context; how are they different from a nurse, a doctor, a physician’s assistant? Are they the same as an NP in the UK, US or other national context? (see https://www.ncbi.nlm.nih.gov/pmc/articles/PMC5020757/). This would be the ideal place to detail numbers of professionals as as is already done.
The section setting the scene on public opinions on nurse practitioners is good, though by conflating all the citations 10-19 together, we lose what might be particularly culturally-bounded about these attitudes. Are they universal attitudes, or are they fed by the particular national culture of Poland in terms of gender and profession ideals? How do they compare with public ideals about nurses/nursing in other cultures or settings, e.g. Asian societies (see http://dx.doi.org/10.1097/JNR.0b013e3181dda76a). Also need to explore why public opinions on NPs, or any other healthcare workers, matter in the first place; how does the way the public feel about healthcare practitioners inform public opinion on healthcare provision or spending more generally, particularly in crisis?
I felt of the 4 research questions, “Has the COVID-19 pandemic turned the spotlight on to NPs?” is too vague, too closed a question, and has already been answered by the introduction. This is also the case for the question “Has the social role of NPs has been socially recognised?” – do you mean HOW has the social role of NPs been socially recognised? And do you mean within the context of the pandemic (which seems to be covered by Q4) or in a broader cultural/historical context (which the introduction already covers). The two questions “What social reactions did NPs face during the pandemic?” and “Has the pandemic influenced the social perception of NPs?” are good, though perhaps “changed” might be stronger than “influenced”.
Are all the cited references relevant to the research?
Those used are mostly relevant, but more supporting references needed, some additions are suggested throughout here.
Is the research design appropriate?
No, this is significantly under-developed. I am unsure why the Qualitative Pretest Interview (QPI) approach was established at the heart of this research design; this is a method for evaluating and editing survey questions in quantitative questionnaires. This is not a standalone methodology, particularly not for robust qualitative research. It is sufficient to say that semi-structured interviews were used as part of a qualitative approach, but the authors need to better establish what epistemology fed their approach, and why qualitative semi-structured interviews are the most suitable means of doing so. The authors mention using Interpretative Phenomenological Analysis (IPA) as their data analysis method in section 2.3; this is more accurately an overarching methodology which governs the whole project, not just the data analysis, and the details about the selection and justification of the IPA approach should be covered in this initial section instead.
Are the methods adequately described?
No, much more clarification needed here. The way the research method design is described is unclear and problematic, possibly through a linguistic issue but this is likely the root of issues with the research design. Rather than a “questionnaire”, what the authors have developed is an “interview schedule” or “topic guide”, if they are doing semi-structured interviews; developing this iteratively with stakeholders and potential participants is a good practice to do, but it is not an application of the QPI approach, merely standard qualitative research practice (as suggested by most introductory guides to qualitative research like Braun & Clarke’s book Successful qualitative research: A practical guide for beginners, 2013)
Despite the fact that Polish law requires no formal ethical approval, attention should still be paid to how the authors ensured an “ethics of care” for their participants in the interviews and in the subsequent outputs (see Antoni & Beer 2019 Research impact as care: re-conceptualizing research impact from an ethics of care perspective; in Fotaki Islam, Antoni Business Ethics and Care in Organizations pp. 172-186.) Despite no national requirements, there is still an international requirement for ethical research practice, for example the International Sociological Association's (ISA) Code of Ethics. This is particularly important given the emotional nature of the situation and lines of questioning. How did the authors ensure the participants were psychologically safe, and anonymous, and free to give their experiences and opinions without fear of harm or repercussion? How did the researchers keep themselves psychologically safe when talking about such themes and lines of questioning?
There is a distinct lack of clarity about recruitment here. Snowball sampling seems to have been the secondary recruitment method, but what about the primary recruitment? The authors state that “Nine hospital nurses were initially recruited and they were asked to provide referrals to other potential subjects” (line 131-2) – were they individually directly approached (selectively sampled)? How were they approached; personally by the author or authors, or self-selecting through an advertisement? If personally recruited, was there an established relationship between the authors and participants to facilitate this, and what was it? Also be more specific about the criteria; was this just Polish hospitals?
There is a bit of confusion throughout the methods section – if just NPs are the focus of the research and just NPs were participants, the authors should avoid using “nurse” interchangeably for “nurse practitioner” for greater specificity.
What language were interviews conducted in? If in Polish, but any quotes and themes in this paper have been translated to English, this should be noted, as it puts an additional analytical lens over the participants’ words since the English-language reader is not reading the “raw” words but how they have been interpreted by the translator.
As mentioned above, IPA is a research methodology rather than an analysis method, so it is suggested this justification for the use of IPA move up earlier in the section. Also need to detail the coding approach, and where codes have come from – see https://journals.sagepub.com/doi/pdf/10.1177/160940690600500107 which, although about thematic analysis rather than IPA, defines coding approaches and their combination well.
Are the results clearly presented?
In table 1, suggest “gender” rather than “sex”, and “male/female” rather than “woman/man” unless the authors specifically asked the participants “what is your sex, man or woman”?
I find the inclusion of such specific information – exact age, gender, length of tenure, specific ward site and employment type – very uncomfortably identifiable to insiders within this system; anyone working in one of these sites could probably work out exactly who, or a short list of potential people, was represented by a specific code. This is not very ethical practice at all and risks harm to the participants as they may not be anonymous. I recommend age ranges (e.g. 38 becomes 30-40), and removing Hospital Ward. Some of the Hospital Ward details could move to Specialisation to give more context, and the authors might still find merit in designating whether someone worked in an Urban or Remote location, or a Public or Private facility, but not down to specific site level.
Theme 1, “NPs’ reactions to the COVID-19 pandemic”, seems to be outside of the domain of the key research question about public views of NPs in the COVID pandemic. It does not fit within the scope of what the paper is trying to do, and I suggest removing this theme from this paper and discussion to focus more fully on the public opinion.
Theme 2 “The impact of the pandemic on NPs’ mental health” generally is of less relevance to the key research question of the paper as exploring the impact of the changed social attitudes towards NPs on their mental health. It would help to refocus this part of the analysis around that particular theme, and have it as the last theme covered; it would then transition nicely into discussing what the real-world implications of all this is and what needs to be addressed properly.
Theme 3 “Social attitudes towards NPs during the pandemic” is very good; however it’s not very clear how this is different to Theme 4 “ The impact of the COVID-19 on the social perception of NPs”. This might be more clear if there was a distinction between a theme of “general social attitudes towards NPs” and then a pandemic-specific one.
All sections would benefit from the identification of an additional layer of sub-themes, reflected in the structure of the sections.
Are the conclusions supported by the results?
These need again to come back to the key research questions of the paper and sit in the scope it is trying to answer. This is fairly well done, with strong links to other international findings. But it misses something around the discussion of the implications of the identified issues. To put it crudely – “so what”? What does Poland risk or gain by having these changed attitudes to NPs as a result of COVID? If there are gains, what is needed to maintain them? If there are risks, how should they be mitigated? Recommend the addition of a “practice and policy implications” section, before Limitations, to make this clear.
Regarding limitations, in the nature of a qualitative study, the insights are not designed to be generalisable to a broader population, just to provide insights that may be useful in other settings and contexts. Lack of generalisability does not mean limited validity. Limited validity might, however, come from having only one method of data collection (just interviews) without triangulating it against other sources like patient interviews or media discourse analysis. The data analysis might also be considered a limitation if each transcript was not coded by both researchers independently to reach consensus/corroboration, rather than just collaborating to decide on codes.
Author Response
Dear Reviewer,
First of all, we would like to express our gratitude for giving us the opportunity to revise and resubmit our paper. We are indebted to you for your valuable suggestions and helpful comments. We hope that this revised paper is more consistent as a result of your willingness to help.
As you will see from this cover letter, we have put a good deal of effort into revising our paper in accordance with your suggestions. We are in agreement with the points you make and we are extremely grateful for pointing these things out and we believe that we have addressed all these points. Should we happen to have missed or misunderstood any vital comment, however, we would be more than happy to promptly rectify this and further revise the article.
Below we detail the changes we have made in accordance with your suggestions and comments. Please note that in the revised manuscript we have taken the liberty of marking the most important changes in colour to facilitate their checking.
At the same time, we would like assure you that, while prior to the original submission the manuscript has been read and copy-edited by a British native speaker, the revised paper has also been read and copy-edited by another British native speaker.
1. Reviewer’s comment:
Thank you for the opportunity to review this manuscript, which uses a qualitative approach to study the experiences and attitudes of nurse practitioners (NPs) in Poland during the COVID-19 pandemic. The paper talks in particular about NPs experiences and perceptions of societal attitudes towards them, and makes recommendations based on their lived experiences and responses for policymakers to improve their quality of working life.
This paper has the potential to make an important contribution to workforce literature, with nurse practitioners a key but understudied group, and Polish healthcare workforce of interest globally, and the examination of societal attitudes is a rich and useful area of focus given the discourse globally about the “healthcare hero” juxtaposed with public and political ‘blindness’ to the working conditions and psychological safety of healthcare workers. The authors suggest that there is a “great shortage of research on the impact of the pandemic on the social perception of NPs…in Poland” (line 102-3) but arguably there has been little examination of this anywhere, and since there are significant post-COVID pressures on healthcare systems and accompanying political and public rhetoric about healthcare workers and their needs and roles around the world (consider public and media discourse around nurse strikes in the UK, Australia and Ireland, rallies in New Zealand, doctors strikes in Malaysia Nigeria and Bolivia in the last 2 years alone), then showing how COVID changed public opinions of NPs in Poland has much broader international insights to provide.
Authors’ response:
Thank you. We appreciate your positive feedback and valuable suggestions on ways to improve our article.
2. Reviewer’s comment:
However, I do think this piece needs substantial work to bring it up to a quality that would be suitable for international scholarly publication. I think a major issue is that the paper does not reflect a robust and well-understood qualitative research practice, and it perhaps tries to focus on too much, with the result that the core focus question of “how did COVID affect people’s perceptions of NPs, how did that affect them, and how do we fix it” gets lost. Re-establishing this core focus (or clarifying if this is not the right interpretation of the study!) throughout the introduction, findings, discussion is required, and the whole piece requires reframing in terms of cementing the scholarly qualitative epistemology and practice that underpins it to promote its robustness. It is also vital that the authors revise substantially their ethical research approach at the publication level, as there are ethical concerns raised regarding the data this paper shares and protection of the participants.
Authors’ response:
We are grateful to you for this remark because it has helped us to clarify the important issues you have raised. Your concerns have persuaded us to revise all methodological aspects of our research, including sampling methods (point 11 of this cover letter), our data analysis (points 8 and 13) and ethical issues related to participants’ data protection (points 10 and 15).
In accordance with your pertinent suggestion we have also focused more on our main research questions, i.e. ‘What social reactions did NPs face during the pandemic?’ and ‘Has the pandemic changed the social perception of NPs?’.
Should we have missed or misunderstood any vital comment, we would be more than happy to promptly rectify this and further revise the article.
At the same time, we are grateful to you for bringing our attention to these important points.
Specific comments:
Does the introduction provide sufficient background and include all relevant references?
3. Reviewer’s comment:
I would suggest opening with a general statement about the importance of the healthcare workforce in the COVID-19 pandemic globally, and providing some basic facts/timeline of the pandemic in Poland in particular to set the scene – this is in the Discussion but needs to be moved much earlier.
Authors’ response:
In accordance with your suggestion we have opened out paper with a short paragraph describing the importance of the healthcare workforce during the COVID-19 pandemic. Following your advice we have also moved paragraphs describing basic facts about the pandemic in Poland from the Discussion to the Introduction. This has been done on pages 1-2 (lines 31-75). The revised passage now reads as follows (the revised part has been marked in colour):
“The coronavirus disease of 2019 (COVID-19) swept rapidly across the world in 2020, infecting more than 600 million people, and was associated with more than six million deaths by the end of 2022 [1]. The pandemic has therefore fundamentally challenged healthcare systems worldwide and has tested their resilience [2]. While the outbreak has indicated the weakest points of the health systems, it has also demonstrated the vital role healthcare professionals (HCPs) play during times of disaster in ensuring a functioning healthcare system and healthy society [3,4]. The pandemic has also highlighted the physical risks and the high levels of psychological stress HCPs experience at work (extremely demanding environments, long working hours, living in constant fear of exposure to the disease, separation from families, social stigmatisation, etc.) [5].
The first case of infection with the recent severe acute respiratory syndrome coronavirus 2 (SARS-CoV-2) in Poland was confirmed by the Polish Minister of Health on 4 March 2020 [6]. While at the beginning of the pandemic the daily number of infections in Poland remained in the dozens, in respect to the Polish legal act on the spread of infection a state of natural disaster was soon declared [7] and on 10 March the Polish government imposed a number of public health restrictions and lockdown-type control measures, including the implementation of social distancing, limits on travel and the closure of all schools, public places, hospitals and long-term care facilities to external visitors. In spite of this, the first death caused by the new virus occurred on 12 March [8] and on 20 March a state of epidemic had been declared in Poland [9].
As in other counties, the pandemic created a situation in which nursing personnel (NPs) had to operate in conditions that represented a threat to their health, influenced their teamwork and the climate of safety, job satisfaction, the perception of management and working conditions and their mental health [5.10,11]. The COVID-19 pandemic also resulted in great suffering among HCPs, including nurses, as from the beginning of the outbreak more than 110,641 nurses became infected in Poland and more than 260 died [12].
Although the pandemic caused chaos in the provision of nursing care in terms of working conditions (disinfections, overalls, masks and shields, gloves, goggles and PPE), and made NPs’ work even more exhausting both physically and emotionally, it also created an opportunity for elevating the public image of the nursing profession [13-15]. The reason for this is that over past two years the critical roles and responsibilities NPs during the pandemic expanded dramatically and came under increased scrutiny. While NPs continued in their roles on the front line of patient care in hospitals and other health care facilities, they were also actively involved in the screening, early diagnosis and continued monitoring of infected patients. They documented patients’ health status and communicated it to other health officials. Finally, NPs were responsible for informing patients about vaccinations and sometimes had to assume the duties of other personnel, i.e. technologists [3,4].
During the first wave of the pandemic especially and as a result of governmental efforts, media coverage and a number of social campaigns, NPs received a great deal of social attention, highlighting their sacrifice, bravery, tremendous efforts, dedication and altruism. This, in turn, increased the visibility of NPs, elicited recognition for their work during the crisis, and highlighted the reason the role of NPs is so important in healthcare. It therefore somehow promoted the image of nursing to the general public.”
We hope this change satisfies your concerns. We are grateful to you for this valuable comment.
4. Reviewer’s comment:
Then focus in on the NPs – what was their crucial role in this? (This is later in the paper, lines 82-90, but needs to come earlier). It would also be helpful to define and detail what constitutes a “nurse practitioner” in the Polish context; how are they different from a nurse, a doctor, a physician’s assistant? Are they the same as an NP in the UK, US or other national context? (see https://www.ncbi.nlm.nih.gov/pmc/articles/PMC5020757/). This would be the ideal place to detail numbers of professionals as as is already done.
Authors’ response:
In accordance with your suggestion we have added a short paragraph describing the critical roles and responsibilities of nursing personnel during the pandemic. This has been done on page 2 (lines 58-69). The revised version now reads as follows (the revised part has been marked in colour):
“Although the pandemic caused chaos in the provision of nursing care in terms of working conditions (disinfections, overalls, masks and shields, gloves, goggles and PPE), and made NPs’ work even more exhausting both physically and emotionally, it also created an opportunity for elevating the public image of the nursing profession [13-15]. The reason for this is that over past two years the critical roles and responsibilities NPs during the pandemic expanded dramatically and came under increased scrutiny. While NPs continued in their roles on the front line of patient care in hospitals and other health care facilities, they were also actively involved in the screening, early diagnosis and continued monitoring of infected patients. They documented patients’ health status and communicated it to other health officials. Finally, NPs were responsible for informing patients about vaccinations and sometimes had to assume the duties of other personnel, i.e. technologists [3,4].”
We admit that we have confused the terms ‘nurse practitioner’ with ‘nurse/nursing personnel’. Since ‘nurse practitioner’ is not legally recognised in Poland, we have revised it throughout the text, i.e. lines 13, 27, 51. We hope this change satisfies your concerns. Again, we are grateful to you for bringing our attention to this point.
5. Reviewer’s comment:
The section setting the scene on public opinions on nurse practitioners is good, though by conflating all the citations 10-19 together, we lose what might be particularly culturally-bounded about these attitudes. Are they universal attitudes, or are they fed by the particular national culture of Poland in terms of gender and profession ideals? How do they compare with public ideals about nurses/nursing in other cultures or settings, e.g. Asian societies (see http://dx.doi.org/10.1097/JNR.0b013e3181dda76a). Also need to explore why public opinions on NPs, or any other healthcare workers, matter in the first place; how does the way the public feel about healthcare practitioners inform public opinion on healthcare provision or spending more generally, particularly in crisis?
Authors’ response:
Having been persuaded by your concerns, we have rewritten the entire paragraph and added suitable references to each stereotype described, though, while we have tried to refer to the work you attached, the link unfortunately failed to open. This has been done on pages 2-3 (lines 94-115). The revised version now reads as follows (the revised part has been marked in colour):
“This social perception of the nursing profession, as defined as a process in which individuals make certain judgements or draw inferences about individuals or groups [25], is profoundly affected by stereotypes, and is formed through a discourses of knowledge and power [26,27] that shapes the public understanding of what is considered to be true and false. Indeed, research shows that nursing is still perceived as a woman’s job and nurses are often imagined via traditional, sentimental stereotypes as selfless, young women [26,28]. They are also ascribed a secondary role and are perceived as lacking in skills and training, with little responsibility, autonomy or decision-making capacity [29-31]. Some also believe that nurses are thought to be failed medical students, i.e. that they are insufficiently intellectually equipped to become doctors [29,32]. This is because the nature of nursing work is often seen as simply caring for and helping patients. Consequently, nurses’ work is perceived as inferior to doctors’ and nurses may be described as doctors’ assistants or servants. Some people believe that nurses’ training is easy and that all hospital nurses do the same kind of work, i.e. provide care for the sick in hospital, administer medication and deal with patients’ hygiene [24,31,33]. While nurses are considered as kind, caring and hardworking they are therefore frequently perceived as less intellectual or autonomous professionals [33,34]. On the other hand, even if NPs are perceived as responsible, skilful, compassionate and kind, people are unaware of such important competences as research, leadership, involvement in decision-making, team work competences and inter-professional communication necessary to modern nursing [23,24,31]. All in all, research suggest that, nursing is frequently imagined as simple, blue-collar work offering low social prestige [23,24,31,32-35].”
Additionally, in accordance with your second comment we have explored the reasons public opinion on NPs is important. This has been done on page 3 (lines 116-119 and 137-144). The revised version now reads (the revised part has been marked in colour):
“This is important because such harmful stereotypes tend to discourage young people from the profession and may give rise to prejudice toward NPs [30]. They may also affect NPs’ mental health, lead to frustration and low self-esteem and a disruption of NPs’ sense of worth, as well as making their job a good deal more difficult [33-36].”
(…)
“At the same time, as the social image of NPs has changed over the decades from that of a servant to patients and subordinate to doctors to a professional caregiver [30], some argue that the social image of NPs may be strengthened during times of disasters such as earthquakes, tsunamis and epidemics [13]. It is also suggested that the SARS-CoV-2 healthcare crisis has clearly stressed the status of the nursing profession and the vital role NPs play in promoting public health; that COVID-19 has provided a unique opportunity to strengthen the public image of NPs and extend their potential to influence health policy and decision-making at both national and global levels [14,15].”
We hope the above explanation satisfies your concerns. Once again, we appreciate your valuable suggestions.
6. Reviewer’s comment:
I felt of the 4 research questions, “Has the COVID-19 pandemic turned the spotlight on to NPs?” is too vague, too closed a question, and has already been answered by the introduction. This is also the case for the question “Has the social role of NPs has been socially recognised?” – do you mean HOW has the social role of NPs been socially recognised? And do you mean within the context of the pandemic (which seems to be covered by Q4) or in a broader cultural/historical context (which the introduction already covers). The two questions “What social reactions did NPs face during the pandemic?” and “Has the pandemic influenced the social perception of NPs?” are good, though perhaps “changed” might be stronger than “influenced”.
Authors’ response:
We are grateful to for this remark. It has helped us to clarify and revise the research questions. Your concerns thus led us to remove two research questions that had already been answered in the Introduction and left only two:
- What social reactions did NPs face during the pandemic?
- Has the pandemic changed the social perception of NPs?
This has been done on page 3 (lines 147-148). We hope this change satisfies your concerns. We are grateful to you for bringing our attention to this point.
7. Reviewer’s comment:
Are all the cited references relevant to the research?
Those used are mostly relevant, but more supporting references needed, some additions are suggested throughout here.
Authors’ response:
In accordance with your suggestion we have updated the literature, which now covers some additional research:
--Chemali, S.; Mari-Sáez, A.; El Bcheraoui, C.; Weishaar H. Health care workers’ experiences during the COVID-19 pandemic: a scoping review. Hum Resour Health. 2022, 20, 27. https://doi.org/10.1186/s12960-022-00724-1.
--Haldane, V.; De Foo, C.; Abdalla, S,M.; Jung, A.S.; Tan, M.; Wu, S.; Chua, A.; Verma, M.; Shrestha, P.; Singh, S.; Perez, T.; Tan S.M.; Bartos, M.; Mabuchi, S.; Bonk, M.; McNab, C.; Werner, G.K.; Panjabi, R.; Nordström, A.; Legido-Quigley H.Health systems resilience in managing the COVID-19 pandemic: lessons from 28 countries. Nat Med. 2021, 27, 964–980. https://doi.org/10.1038/s41591-021-01381-y
--Fereday, J.; Muir-Cochrane, E.C. Demonstrating rigor using thematic analysis: A hybrid approach of inductive and deductive coding and theme development. Int J Qual Methods. 2006, 5(1), 80-92.
--Feldermann, S.K.; Hiebl, M.R. Using quotations from nonEnglish interviews in accounting research. Qualitative Research in Accounting & Management. 2019, 17(2), 229–262.
-- Van Nes, F.; Abma, T.; Jonsson, H.; Deeg, D. Language differences in qualitative research: Is meaning lost in translation? Eur J Ageing. 2010, 7(4), 313–316.
--Fawaz, M.; Anshasi, H.; Samaha, A. Nurses at the front line of COVID-19: Roles, responsibilities, risks, and rights. Am J Trop Med Hyg. 2020, 103(4), 1341–1342.
--Kako J, Kajiwara K. Scoping review: What is the role of nurses in the era of the global COVID-19 pandemic? J Palliat Med. 2020, 23(12), 1566-1567.
8. Reviewer’s comment:
Is the research design appropriate?
No, this is significantly under-developed. I am unsure why the Qualitative Pretest Interview (QPI) approach was established at the heart of this research design; this is a method for evaluating and editing survey questions in quantitative questionnaires. This is not a standalone methodology, particularly not for robust qualitative research. It is sufficient to say that semi-structured interviews were used as part of a qualitative approach, but the authors need to better establish what epistemology fed their approach, and why qualitative semi-structured interviews are the most suitable means of doing so. The authors mention using Interpretative Phenomenological Analysis (IPA) as their data analysis method in section 2.3; this is more accurately an overarching methodology which governs the whole project, not just the data analysis, and the details about the selection and justification of the IPA approach should be covered in this initial section instead.
Authors’ response:
We are indebted to you for this valuable observation. Of course, we are aware that the QPI approach is not a (qualitative) research method but an approach for evaluating and editing interview questionnaires. We agree that this is not an essential part of methodology, so we have removed this part from the text.
More importantly, following your advice we have elaborated our justification for using qualitative semi-structured interviews. We therefore explain, that the rationale for choosing qualitative research was that there is a scarcity of previous work on the topic and such methodology is valuable in giving voice to those whose views are rarely heard. It also facilitates an illumination of their experiences. This has been done on page 4 (lines 155-157). The revised version now reads as follows (the revised part has been marked in colour):
“Since there is a scarcity of research on this topic in Poland, and the research aimed at giving voice to NPs whose views are rarely heard the study has been designed as a qualitative research project [42,43].”
In accordance with your pertinent suggestion we have also moved the description of Interpretative Phenomenological Analysis (IPA) from section 2.3. to the initial section 2.1. Study design (page 4, lines 172-181).
We hope this additional explanation satisfies your concerns. Again, we are grateful to you for bringing this point to our attention.
9. Reviewer’s comment:
Are the methods adequately described?
No, much more clarification needed here. The way the research method design is described is unclear and problematic, possibly through a linguistic issue but this is likely the root of issues with the research design. Rather than a “questionnaire”, what the authors have developed is an “interview schedule” or “topic guide”, if they are doing semi-structured interviews; developing this iteratively with stakeholders and potential participants is a good practice to do, but it is not an application of the QPI approach, merely standard qualitative research practice (as suggested by most introductory guides to qualitative research like Braun & Clarke’s book Successful qualitative research: A practical guide for beginners, 2013)
Authors’ response:
In accordance with your recommendation we have replaced the term ‘questionnaire’ with the more appropriate term ‘interview schedule’ (page 4, lines 159, 164 and 170). As for the QPI approach, as noted in point 8 of this cover letter, following your advice we have removed reference to this approach for evaluating and editing interview questionnaires.
We hope the above explanation satisfies your concerns regarding this matter.
10. Reviewer’s comment:
Despite the fact that Polish law requires no formal ethical approval, attention should still be paid to how the authors ensured an “ethics of care” for their participants in the interviews and in the subsequent outputs (see Antoni & Beer 2019 Research impact as care: re-conceptualizing research impact from an ethics of care perspective; in Fotaki Islam, Antoni Business Ethics and Care in Organizations pp. 172-186.) Despite no national requirements, there is still an international requirement for ethical research practice, for example the International Sociological Association's (ISA) Code of Ethics. This is particularly important given the emotional nature of the situation and lines of questioning. How did the authors ensure the participants were psychologically safe, and anonymous, and free to give their experiences and opinions without fear of harm or repercussion? How did the researchers keep themselves psychologically safe when talking about such themes and lines of questioning?
Authors’ response:
Having been persuaded by your concerns, we have added section 2.3. Ethical Issues, where we describe the ethical issues related to our research. We have added the explanation that all respondents were informed about the voluntary, anonymous and confidential nature of the study and about the possibility to withdraw from the interview at any given moment and/or choose to withhold information on their personal circumstances. They were also informed that all identifying information would be redacted from the records of the interviews and they would be stored in a restricted area with limited access. We then explained that all participants were informed that, if they experienced any emotional distress while recalling past events, a hospital psychologist was available for counselling and that they were at liberty to take their time to collect themselves and/or decide whether to continue the interview. Finally, we declare that informed consent was obtained from all NPs enrolled in our study and that only after that they were scheduled to be interviewed. This has been done on pages 4-5 (lines 201-217). The revised version now reads as follows (the revised part has been marked in colour):
“2.3. Ethical issues
The study was performed in accordance with the principles of the Declaration of Helsinki. In accordance with Polish law and Good Clinical Practice on research involving human subjects this study required no revision by an ethics committee.
NPs received an invitation letter and were informed about the study’s purpose, as well as the voluntary, anonymous and confidential nature of the study, and about the possibility of withdrawing from the interview at any given moment and/or not disclosing information regarding their personal circumstances. Respondents were also informed that identifying information would be redacted from records of the interviews and they would be stored in a secure place with restricted access. Participants were informed that, should they experience any emotional distress while recalling past events, a hospital psychologist was available for counselling and that they could take time to collect themselves and/or decide whether to continue the interview.
After informed consent was obtained from all respondents enrolled in the study they were scheduled to be interviewed. To ensure that the participants were psychologically safe and anonymous, all interviews took place in private, isolated rooms. All interviews ended without any requests for further assistance”
We hope the above explanation satisfies your concerns. We are grateful to you for bringing our attention to this important point.
11. Reviewer’s comment:
There is a distinct lack of clarity about recruitment here. Snowball sampling seems to have been the secondary recruitment method, but what about the primary recruitment? The authors state that “Nine hospital nurses were initially recruited and they were asked to provide referrals to other potential subjects” (line 131-2) – were they individually directly approached (selectively sampled)? How were they approached; personally by the author or authors, or self-selecting through an advertisement? If personally recruited, was there an established relationship between the authors and participants to facilitate this, and what was it? Also be more specific about the criteria; was this just Polish hospitals?
Authors’ response:
We are indebted to you for this remark. It has helped us to clarify the recruitment process. On page 4 (lines 187-195) we have provided the explanation that the first nine nurses were recruited through an advertisement that was posted on an online platform for nurses. After making personal contact with the first author [KW], NPs gave informed consent and were scheduled to be interviewed. The revised version now reads as follows (the revised part has been marked in colour):
“Participants were identified via an advertisement that was posted on an online platform for nurses. This enabled us to identify nine hospital nurses. In order ensure validity of the results, however, recruitment of NPs continued until thematic saturation occurred. To achieve that, a non-probability snowball sampling method was also used [46]. After making personal contact with the first author [KW] NPs who were scheduled to be interviewed were therefore asked to provide referrals to other potential subjects. This in turn helped to recruit six additional nurses. The recruitment process thus continued until thematic saturation was achieved [47]. All in all, fifteen nurses responded and agreed to an interview.”
While describing the criteria for inclusion on page 4 (line 184) we have also added the information that the participants were chosen from among nurses who worked in Polish hospitals.
“The participants were chosen from among nurses who worked in Polish hospital, were directly involved in hospital care during the COVID-19 pandemic and were willing to participate in the study.”
We hope the above explanation satisfies your concerns. We are grateful for to you for bringing our attention to this important point.
12. Reviewer’s comment:
There is a bit of confusion throughout the methods section – if just NPs are the focus of the research and just NPs were participants, the authors should avoid using “nurse” interchangeably for “nurse practitioner” for greater specificity.
Authors’ response:
As explained in point 4. of this cover letter, as we have confused the term ‘nurse practitioner’ with ‘nurse/nursing personnel’ we have revised it accordingly. There are thus no longer any references to ‘nurse practitioners’. Again, we are grateful for your bringing this to our attention.
13. Reviewer’s comment:
What language were interviews conducted in? If in Polish, but any quotes and themes in this paper have been translated to English, this should be noted, as it puts an additional analytical lens over the participants’ words since the English-language reader is not reading the “raw” words but how they have been interpreted by the translator.
As mentioned above, IPA is a research methodology rather than an analysis method, so it is suggested this justification for the use of IPA move up earlier in the section. Also need to detail the coding approach, and where codes have come from – see https://journals.sagepub.com/doi/pdf/10.1177/160940690600500107 which, although about thematic analysis rather than IPA, defines coding approaches and their combination well.
Authors’ response:
We are grateful to for this remark. It has helped us to clarify that all interviews were conducted in Polish and all quotations and themes were translated into English during the data analysis stage and reported with the help of bilingual translator [Feldermann and Hiebl 2019]. We are aware that every language reflects important nuances that are inherent to participants’ experiences and situational contexts and that translation of participants’ responses from one language to another may disrupt theses nuances and meaning they describe may be lost. Every translation involves interpretation on the part of translators [Nes et al., 2010]. For that reason we have added this caution as yet another limitation. This has been done on page 15 (lines 702-708). We hope the above explanation satisfies your concerns.
In accordance with your suggestion we have moved the justification for the use of our research method (IPA) to the earlier section 2.1. Study design (page 4, lines 172-178).
Are the results clearly presented?
14. Reviewer’s comment:
In table 1, suggest “gender” rather than “sex”, and “male/female” rather than “woman/man” unless the authors specifically asked the participants “what is your sex, man or woman”?
Authors’ response:
In accordance with your suggestion in Table 1 (page 5) we have changed ‘sex’ to ‘gender’, and ‘woman/man’ to ‘male/female’.
15. Reviewer’s comment:
I find the inclusion of such specific information – exact age, gender, length of tenure, specific ward site and employment type – very uncomfortably identifiable to insiders within this system; anyone working in one of these sites could probably work out exactly who, or a short list of potential people, was represented by a specific code. This is not very ethical practice at all and risks harm to the participants as they may not be anonymous. I recommend age ranges (e.g. 38 becomes 30-40), and removing Hospital Ward. Some of the Hospital Ward details could move to Specialisation to give more context, and the authors might still find merit in designating whether someone worked in an Urban or Remote location, or a Public or Private facility, but not down to specific site level.
Authors’ response:
We are grateful to you for this remark. It has helped us to comply better with the ethical practice of assuring the participants of their confidentiality. We are persuaded by your concerns and, instead of giving the exact age, we have used the age ranges (18-24, 25-29, 30-34, 35-39, 40-44, 45-49, 50-54, 55-59). In accordance with your suggestion we have also removed the hospital ward details and have only described public and private facilities (all participants worked in an urban location). This has been done on pages 5-6. We hope this revision satisfies your ethical concerns regarding data protection. We are grateful for to you for bringing this to our attention.
16. Reviewer’s comment
Theme 1, “NPs’ reactions to the COVID-19 pandemic”, seems to be outside of the domain of the key research question about public views of NPs in the COVID pandemic. It does not fit within the scope of what the paper is trying to do, and I suggest removing this theme from this paper and discussion to focus more fully on the public opinion.
Authors’ response:
In accordance with your suggestion we have removed this theme from both the paper and discussion, and focused only on nurses’ experiences on the way the COVID-19 pandemic influenced the social image of nursing.
17. Reviewer’s comment
Theme 2 “The impact of the pandemic on NPs’ mental health” generally is of less relevance to the key research question of the paper as exploring the impact of the changed social attitudes towards NPs on their mental health. It would help to refocus this part of the analysis around that particular theme, and have it as the last theme covered; it would then transition nicely into discussing what the real-world implications of all this is and what needs to be addressed properly.
Authors’ response:
We are grateful to you for this remark. As suggested we have moved Theme 2 to the end.
18. Reviewer’s comment
Theme 3 “Social attitudes towards NPs during the pandemic” is very good; however it’s not very clear how this is different to Theme 4 “ The impact of the COVID-19 on the social perception of NPs”. This might be more clear if there was a distinction between a theme of “general social attitudes towards NPs” and then a pandemic-specific one.
All sections would benefit from the identification of an additional layer of sub-themes, reflected in the structure of the sections.
Authors’ response:
In accordance with your suggestion we have renamed theme 4 (now 2) and it is called ‘NPs’ perception of the impact of the pandemic on the social image of nursing’. We have also included the additional sub-themes:
3.2.1. Social attitudes towards NPs during the pandemic (3.2.1.1. NPs as heroes: pride, respect and admiration; 3.2.1.2. Ambivalence: between indifference and suspicion; 3.2.1.3. NPs as frauds: prejudice, stigmatisation and discrimination),
3.2.2. NPs’ perception of the impact of the pandemic on the social image of nursing profession (3.2.2.1. Spotlight on nursing; 3.2.2.2. The role of the (social)media; 3.2.2.3. Social image of NPs before COVID-19; 3.2.2.4. Social image of NPs after COVID-19; 3.2.2.5. COVID-19 and NPs’ professional identity),
3.2.3. The impact of the pandemic on NPs’ mental health (3.2.3.1. COIVD-19 and disrupted lives; 3.2.3.2. COIVD-19 and disrupted work; 3.2.3.3. COIVD-19 and emotional distress)
We hope this change satisfies your concerns. We are grateful to you for this valuable comment.
19. Reviewer’s comment:
Are the conclusions supported by the results?
These need again to come back to the key research questions of the paper and sit in the scope it is trying to answer. This is fairly well done, with strong links to other international findings. But it misses something around the discussion of the implications of the identified issues. To put it crudely – “so what”? What does Poland risk or gain by having these changed attitudes to NPs as a result of COVID? If there are gains, what is needed to maintain them? If there are risks, how should they be mitigated? Recommend the addition of a “practice and policy implications” section, before Limitations, to make this clear.
Authors’ response:
In accordance with your suggestion we have rewritten the conclusion, so that it now reflects more on our research questions: ‘What social reactions did NPs face during the pandemic?’ and ‘Has the pandemic changed the social perception of NPs?’ This has been done on page 14 (lines 651-668). The revised version now reads as follows (the revised part has been marked in colour):
“5. Conclusions
This study shows that although most respondents experienced positive reactions and social gratitude for their service and involvement in the fight against the pandemic, still many NPs faced prejudice, stigmatisation or discrimination, which made them feel undervalued.
At the same time, our findings confirm that the COVID-19 outbreak created an opportunity to strengthen the social image of NPs. Many participants, in fact, believed that the pandemic has helped to overcome harmful stereotypes of nurses and disseminate the image of a NPS as hardworking, well-educated, highly skilled and independent professionals. This in turn, improved respondents’ self-esteem and professional identity.
This study, however, also shows that, since the government, the media and patients often highlighted the gendered stereotype of a nurse as a caring and empathetic professional, the pandemic might further undermine NPs’ professionalism and divert social attention from the knowledge, education and skills required to become a competent nurse. While all NPs enrolled in this study acknowledged that the pandemic increased the social visibility of NPs, who have been recognised as essential healthcare workers, they were also frustrated that in times of healthcare disasters they still had to struggle for social, professional and economic recognition.”
In accordance with your suggestion before Limitations we have also added another section: ‘6. Implications for policy and practice’, where we suggest a systemic approach should be undertaken in order to promote the status of nurses and improve their working conditions and psychological safety. This has been done on page 14 (lines 670-688). The revised version now reads as follows (the revised part has been marked in colour):
“6. Implications for policy and practice
As all respondents highlighted personal risk, lack of adequate PPE, shortages of skilled staff and low nurse-to-patient ratios (particularly in critical care), this study also suggests that social admiration for NPs should not distract our attention from decades of neglecting health services, failing to prepare adequately for epidemic crises and neglecting to foster the social prestige of the nursing profession. It therefore stresses that society and policy-makers have a duty to undertake a systematic approach to improving the organisation of healthcare, increase NPs’ safety by providing them with a safe working environment, preparing them better for the next health crisis and rewarding NPs for their sacrifices.
1. While the COVID-19 outbreak has helped raise social awareness of the role of nurses in the healthcare system, there is an urgent need for public communication and the promotion of a more truthful image of nursing in society.
2. As working during the outbreak was physically and emotionally challenging, special attention must be paid to HCPs’ safety, and they must be provided with both the necessary personal protective equipment and mental health support.
3. NPs and HCPs in general should be better prepared for future global health crises and medical disasters, emergency decision-making, coping and leadership during a crisis by the medical curricula.”
We hope this change satisfies your concerns. We are grateful to you for this valuable suggestion.
20. Reviewer’s comment:
Regarding limitations, in the nature of a qualitative study, the insights are not designed to be generalisable to a broader population, just to provide insights that may be useful in other settings and contexts. Lack of generalisability does not mean limited validity. Limited validity might, however, come from having only one method of data collection (just interviews) without triangulating it against other sources like patient interviews or media discourse analysis. The data analysis might also be considered a limitation if each transcript was not coded by both researchers independently to reach consensus/corroboration, rather than just collaborating to decide on codes.
Authors’ response:
We are indebted to you for this valuable comment. It has helped us to describe the additional limitations of our study you mentioned. As we are aware that qualitative studies are not designed to be generalisable to a broader population, we are persuaded by your apt comment and have removed this point from the Limitations section. This has been done on pages 14-15 (lines 692-708). The revised version now reads as follows (the revised part has been marked in colour):
“Although to the best of our knowledge this is the first study to explore NPs’ experiences during the COVID-19 pandemic in Poland, it is not without limitations. Firstly, even though thematic saturation was achieved, the study sample was relatively small, only fifteen NPs being interviewed. Secondly, because NPs from only one Polish city were interviewed the study is geographically limited. Thirdly, female participants predominated (F:M 13:2), which limits the results’ transferability to male NPs. This imbalance is nevertheless representative of the female predominance within the nursing profession in Poland. A further limitation is that only one method of data collection was used without triangulating it against other sources, i.e. patient interviews or media discourse analysis. The data analysis might also be considered a limitation, as researchers did not code the transcripts independently but just collaborated to decide on codes. Finally, this study’s involving non-English-speaking participants means that all interviews were conducted in Polish and all quotes and themes were translated to English with the help of bilingual translators. Since every language reflects important nuances that are inherent in participants’ experiences and situational context, translation of participants’ quotes from one language to another may disrupt theses nuances and some meanings may be lost. Every translation involves the risk of misinterpretation on the part of researchers [70].
Despite these limitations, however, the benefits of this study should also be acknowledged. Most importantly, since there is a limited amount of prior research on the topic, this study narrows the gap in the literature regarding the experiences of NPs during the COVID-19 pandemic in Poland.”
We hope this change satisfies your concerns. We are grateful to you for this valuable comment.
Again, we wish to thank you for all your valuable suggestions and willingness to help us in improving our manuscript.

Round 2
Reviewer 2 Report
Thank you very much to the authors for engaging in such a rigorous overhaul of their manuscript following the first round of feedback. The resulting paper is much stronger, and I think almost ready for publication after a few minor, mostly textual, revisions.
1. The use of the acronym "NP" throughout is now not necessary as the authors are using "nurse" rather than "nurse practitioner". Suggest doing a blanket search for NP/NPs throughout the entire text and changing simply to nurse/nurses.
2. Introduction Line 43-48: this is a very long sentence; suggest breaking this up into a separate sentence after "measures" (line 46), then start the new one "This included..."
3. Line 94 "This social perception of the nursing profession" - here it needs to be clearer that the specific Polish data about views of nurses cited above (line 89-93) reflects other trends in attitudes worldwide; the literature being drawn on in this section comes from all over the world, though the text seems to insinuate that these are only Polish attitudes. Regarding specific Polish attitudes, there was a comment last time asking whether there were any specific attitudes within Polish traditional culture about nursing, particularly as a feminized role and around women's work generally - could the authors consider a brief sentence or two around this?
4. In 2.1, the sentence beginning "Since there is a scarcity of research" (lines 155-157) is very difficult to understand. Is the sense meant to be that while there is research on this topic in Poland (as evidenced by the cited surveys the authors have included), there is a lack of research which gives voice to the nurses themselves and their lived experiences, and that's why this qualitative study was deemed necessary?
5. Line 158-9: should this be "After a thorough analysis of academic literature"?
6. Line 188 - it is still important to be clear that the participating nurses self-selected, and voluntarily contacted the researchers to be involved in the study. Instead of "This enabled us to identify nine hospital nurses", suggest "nine hospital nurses initially responded to this advertisement and volunteered to participate".
7. Section 2.3 - suggest including a reference supporting/justifying the ethical care approach used by the authors to ensure the psychological safety of the participants and themselves; suggest McCosker, Barnard & Gerber 2001 https://doi.org/10.17169/fqs-2.1.983.
8. p.7 figure 1 - check the spelling of COVID is correct and consistent across the figure
9. Ensure consistency throughout, choose one of either Covid-19 or COVID-19 (WHO prefers the first option) but do not use interchangeably
10. Put the Conclusion (5) last after the Implications (6) and Limitations (7)
11. Reference 1, is there a more official equivalent resource that could be used, perhaps from the WHO?
Author Response
Dear Reviewer,
first of all, we would like to express our gratitude for giving us another opportunity to further revise and resubmit our paper. We are also indebted for all valuable suggestions and helpful comments. We have been convinced by all your additional arguments and we believe that we have answered all the questions. Thus, we hope that this revised paper is more consistent owing to your willingness to help. Still, should it happen that we have missed and/or misunderstood any vital comment, we would be more than happy to promptly rectify this and further revise our article.
Below we detail the changes we have made in accordance with your suggestions and comments. Please note that while all previous changes are still marked in colour red, in this revised manuscript we have taken the liberty of marking all additional changes in colour blue to facilitate their checking.
Sincerely
Jan Domaradzki
Reviewer’s comment:
Thank you very much to the authors for engaging in such a rigorous overhaul of their manuscript following the first round of feedback. The resulting paper is much stronger, and I think almost ready for publication after a few minor, mostly textual, revisions.
1. Reviewer’s comment:
The use of the acronym "NP" throughout is now not necessary as the authors are using "nurse" rather than "nurse practitioner". Suggest doing a blanket search for NP/NPs throughout the entire text and changing simply to nurse/nurses.
Authors’ response:
In accordance with your valuable suggestion we have removed the acronym “NP/NPs” throughout the text and changed it to “nurse/nurses” (all these changes have been marked in colour blue).
2. Reviewer’s comment:
Introduction Line 43-48: this is a very long sentence; suggest breaking this up into a separate sentence after "measures" (line 46), then start the new one "This included..."
Authors’ response:
We are grateful to for this remark. It has helped us to make the sentence more readable. Thus, in accordance with your suggestion we have broken this long sentence into two separate sentences. This has been done on page 2 (lines 43-48). We hope this change satisfies your concerns. The revised passage now reads as follows (the revised part has been marked in colour blue):
“While at the beginning of the pandemic the daily number of infections in Poland remained in the dozens, in respect to the Polish legal act on the spread of infection a state of natural disaster was soon declared [7] and on 10 March the Polish government imposed a number of public health restrictions and lockdown-type control measures. This included the implementation of social distancing, limits on travel and the closure of all schools, public places, hospitals and long-term care facilities to external visitors.”
Again, we are grateful to you for this valuable comment.
3. Reviewer’s comment:
Line 94 "This social perception of the nursing profession" - here it needs to be clearer that the specific Polish data about views of nurses cited above (line 89-93) reflects other trends in attitudes worldwide; the literature being drawn on in this section comes from all over the world, though the text seems to insinuate that these are only Polish attitudes. Regarding specific Polish attitudes, there was a comment last time asking whether there were any specific attitudes within Polish traditional culture about nursing, particularly as a feminized role and around women's work generally - could the authors consider a brief sentence or two around this?
Authors’ response:
Having been persuaded by your concerns, we have rewritten the entire paragraph. Thus, we begin with the description of the most common cultural stereotypes on nursing in Poland (lines 98-109) and then we stress that some of these stereotypes are also present in other countries (lines 110-120). The revised version now reads as follows (the revised part has been marked in blue colour):
“This social perception of the nursing profession, as defined as a process in which individuals make certain judgements or draw inferences about individuals or groups [25], is profoundly affected by stereotypes, and is formed through a discourses of knowledge and power [26,27] that shapes the public understanding of what is considered to be true and false. Indeed, research shows that nursing is still perceived in Poland as a woman’s job and nurses are often imagined via traditional, sentimental stereotypes as selfless, young women [26,28,29]. Moreover, due to historical and cultural reasons they are often ascribed a secondary role and are perceived as lacking skills and training, with little responsibility, autonomy or decision-making capacity [23,24,28,30]. Some also believe that nurses are thought to be failed medical students, i.e. that they are insufficiently intellectually equipped to become doctors [24,28]. This is because the nature of nursing work is often seen as simply caring for and helping patients. Consequently, nurses’ work is perceived as inferior to doctors’ and nurses are perceived as doctors’ assistants or servants [23,24,28]. On the other hand, male nurses are often labelled as unmanly or gay and have to challenge the stereotypical view that they are unsuitable caregivers [26,29].
However, these stereotypes are also present in other countries. Numerus studies report that while nurses are considered as kind, caring and hardworking they are also frequently perceived as less intellectual or autonomous professionals [31-33]. Moreover, people often believe that nurses’ training is easy and that all hospital nurses do the same kind of work, i.e. provide care for the sick in hospital, administer medication and deal with patients’ hygiene [31-34]. On the other hand, even if nurses are perceived as responsible, skilful, compassionate and kind, people are unaware of such important competences as research, leadership, involvement in decision-making, team work competences and inter-professional communication necessary to modern nursing [31,33]. All in all, research suggest that, nursing is frequently imagined as simple, blue-collar work offering low social prestige [31-35].”
We hope the above change satisfies your concerns. Once again, we appreciate your valuable comment.
4. Reviewer’s comment:
In 2.1, the sentence beginning "Since there is a scarcity of research" (lines 155-157) is very difficult to understand. Is the sense meant to be that while there is research on this topic in Poland (as evidenced by the cited surveys the authors have included), there is a lack of research which gives voice to the nurses themselves and their lived experiences, and that's why this qualitative study was deemed necessary?
Authors’ response:
In accordance with your suggestion we have reformulated the sentence to make it more readable. Thus, it now reads as follows (the revised part has been marked in colour blue):
“Since there is a scarcity of research on the impact of the COVID-19 on the social perception of nurses in Poland, and because research rarely give voice to the nurses themselves and their lived experiences this study has been designed as a qualitative research [42,43].”
This has been done on page 4 (lines 159-162). We hope this change satisfies your concerns. Again, we are grateful to you for bringing our attention to this point.
5. Reviewer’s comment:
Line 158-9: should this be "After a thorough analysis of academic literature"?
Authors’ response:
Thank you. Of course, you are right. It has been revised accordingly.
6. Reviewer’s comment:
Line 188 - it is still important to be clear that the participating nurses self-selected, and voluntarily contacted the researchers to be involved in the study. Instead of "This enabled us to identify nine hospital nurses", suggest "nine hospital nurses initially responded to this advertisement and volunteered to participate".
Authors’ response:
In accordance with your apt suggestion we have revised this sentence accordingly. Thus, after revision it now reads as follows (the revised part has been marked in colour):
“Participants were identified via an advertisement that was posted on an online platform for nurses. Nine hospital nurses initially responded to this advertisement and volunteered to participate. In order ensure validity of the results, however, recruitment of nurses continued until thematic saturation occurred. To achieve that, a non-probability snowball sampling method was also used [46].”
This has been done on page 4 (lines 193-194). We are grateful to you for this valuable suggestion.
7. Reviewer’s comment:
Section 2.3 - suggest including a reference supporting/justifying the ethical care approach used by the authors to ensure the psychological safety of the participants and themselves; suggest McCosker, Barnard & Gerber 2001 https://doi.org/10.17169/fqs-2.1.983.
Authors’ response:
We are grateful to you for bringing our attention to this paper. Thus, in accordance with your suggestion we have added this reference:
- McCosker, H.; Barnard, A.; Gerber, R. Undertaking sensitive research: Issues and strategies for meeting the safety needs of all participants. Forum Qual Sozialforschung/Forum Qual Soc Res. 2001, 2(1). https://doi.org/10.17169/fqs-2.1.983.
This has been done on page 5 (line 221) and in References (page 17, lines 836-837).
8. Reviewer’s comment:
p.7 figure 1 - check the spelling of COVID is correct and consistent across the figure
Authors’ response:
While we have revised the spelling of COVID across the figure in accordance with your suggestion we have also unified its spelling throughout the text.
9. Reviewer’s comment:
Ensure consistency throughout, choose one of either Covid-19 or COVID-19 (WHO prefers the first option) but do not use interchangeably
Authors’ response:
As mentioned above, we have revised the spelling of the Coronavirus Disease 2019 throughout the text. At the same time, after checking it on several WHO’s web pages where it is suggested that when writing about the coronavirus there is a preference for ‘COVID-19’ rather than ‘Covid-19’ (i.e. https://www.who.int/emergencies/diseases/novel-coronavirus-2019; https://www.who.int/publications/i/item/WHO-2019-nCoV-Surveillance_Case_Definition-2022.1). Consequently, we have chosen ‘COVID-19’. We hope the above explanation satisfies your concerns. Once again, we appreciate your valuable suggestions.
10. Reviewer’s comment:
Put the Conclusion (5) last after the Implications (6) and Limitations (7)
Authors’ response:
In accordance with your suggestion we have put the Conclusion (7) as a last section, just after the Implications (5) and Limitations (6)
11. Reviewer’s comment:
Reference 1, is there a more official equivalent resource that could be used, perhaps from the WHO?
Authors’ response:
Having been persuaded by your concerns, we have replaced previous reference 1 with more official equivalent resource:
1. WHO Coronavirus (COVID-19) Dashboard. Available from: https://covid19.who.int (accessed on 3.02.2023).
We hope this change satisfies your concerns. We are grateful to you for bringing our attention to this point.
All in all, again, we wish to thank you for all your additional suggestions and willingness to help us yet another time in improving our manuscript.
